# Learning to Solve an Order Fulfillment Problem in Milliseconds with Edge-Feature-Embedded Graph Attention

## Abstract

The order fulfillment problem is one of the fundamental combinatorial optimization problems in supply chain management and it is required to be solved in real-time for modern online retailing. Such a problem is computationally hard to address by exact mathematical programming methods. In this paper, we propose a machine learning method to solve it in milliseconds by formulating a tripartite graph and learning the best assignment policy through the proposed edge-feature-embedded graph attention mechanism. The edge-feature-embedded graph attention considers the high-dimensional edge features and accounts for the heterogeneous information, which are important characteristics of the studied optimization problem. The model is also size-invariant for problem instances of any scale, and it can address cases that are completely unseen during training. Experiments show that our model substantially outperforms the baseline heuristic method in optimality. The online inference time is milliseconds, which is thousands of times faster than the exact mathematical programming methods.

## 1 Introduction

The current decade has witnessed rapid development of online retailing, which generally has a very large operation scale and poses new challenges to supply chain management. One of the most important problems in the online retailing is the real-time order fulfillment decision, which aims to find the most cost-effective order-warehouse assignment after the customers place the orders online. The decisions are required to be made in real time, because the fulfillment system must complete hundreds of orders per second (Acimovic & Graves, 2015; Xu et al., 2009). A good fulfillment policy could reduce expenses by millions of dollars for large online retailers such as Alibaba and Amazon.

The order fulfillment problem in its most general form is as follows: There are a number of orders and a number of warehouses. Each order contains several items, which are at different inventory levels and have different sale-forbidden[1] dates in these warehouses. Any warehouse can be assigned to serve any order, incurring delivery transportation cost and inventory management cost that vary depending on the locations of the orders and the types of items. An order can be split into multiple packages that are committed by different warehouses due to practical constraints, such as the shortage of inventory. A group of identical items in any order is defined as a *suborder*, which is the smallest assignment unit in this problem. The retailer seeks to find the optimal suborder-warehouse assignment policy that minimizes the overall cost in the order fulfillment.

The order fulfillment problem falls into the category of generalized assignment problems, which is an NP-hard combinatorial optimization (Kundakcioglu & Alizamir, 2009) (An extra note on generalized assignment problem is listed in the Appendix A.1). Traditional approaches to this problem can be divided into exact methods and heuristics. The exact methods guarantee finding optimal solutions by using a branch-and-bound search (B&B) with a mixed-integer programming formulation (MIP), but they generally require a long time to solve and are even computationally intractable for large-scale cases. Because the order fulfillment is a real-time decision task in the online retailing,

---

[1]The sale-forbidden date is usually months ahead of the expiration date. Any item going beyond the sale-forbidden date can no longer be sold.

the exact methods are unable to meet the allowed maximum computation time (tens of milliseconds). The heuristics are typically a set of specialized rules and are currently used in the industry because of the low computational cost(Sethanan & Pitakaso, 2016; D'Ambrosio et al., 2020). As a trade-off, the optimality is sacrificed and once the conditions of the problem change, the rules must be re-designed carefully, which costs a lot of manpower.

Recently, machine learning (ML) has made much progress on solving combinatorial optimization problems (Bengio et al., 2018), e.g., using supervised or reinforcement learning to solve the travelling salesman problem (TSP) and the vehicle routing problem (VRP) in an end-to-end approach (Vinyals et al., 2015; Kool & Welling, 2018; Lu et al., 2020). According to the research, ML methods can achieve stronger policies than heuristics without requiring specialized human knowledge. The computational cost of ML inference is extremely low. The ML methods are also self-adaptive to the changes on the problem conditions, because they are end-to-end methods from the original data to the solutions. This progress motivates us to explore whether order fulfillment in the supply chain industry can be addressed by ML methods.

However, the nature of the order fulfillment problem is size-variant and contains heterogeneous information. Different scales of problem instances have varying numbers of orders, items and warehouses. How can we properly transform the combinatorial optimization problem into the input/output of an ML model? How can we design an ML model that can effectively solve the varying scales of problem instances? These questions pose great challenges to our work.

In this paper, we will for the first time apply ML to solve the real-time order fulfillment decision problem by developing a graph mapping solution framework. A size-invariant graph model is developed to learn the hidden information on the heterogeneous graph. A novel graph attention mechanism is proposed to consider the high-dimensional edge features in the represented optimization problem. The specific contributions are threefold: (1) formulate the real-time order fulfillment decision problem with constraints of the modern online retailing and supply chain industry; (2) propose a tripartite heterogeneous graph representation of the order fulfillment problem and develop an ML-based solution framework; and (3) propose size-invariant and edge-feature-embedded graph attention to learn the hidden information in the heterogeneous graph.

## 2 RELATED WORK AND GAPS

Generally, applying ML on combinatorial optimization can be divided into two approaches. The first approach is to improve the branch-and-bound algorithm in the MIP formulation by using ML for variable branching (Alvarez et al., 2017), node selection (He et al., 2014) and so on. Although these methods can guarantee exact solutions by relying on the branch-and-bound approach, the acceleration is limited. Our focus is on the other type of method, the end-to-end learning approach (Bengio et al., 2018), which learns the optimization policy directly from the problem instances.

The end-to-end ML models for combinatorial optimization require careful design, because both of the problem input and output vary in size (Bengio et al., 2018). This circumstance means that the ML model must be able to address the data sizes that are completely unseen during training. To tackle this problem, the pointer network (Ptr-Net) is introduced based on the sequence-to-sequence model (Vinyals et al., 2015; Bello et al., 2017). The Ptr-Net directly uses an attention mechanism as a "pointer" to select a member of the input sequence as the output, regardless of the varying output dictionary sizes that depend on the problem instances. According to the experiments, the model performs well on the convex hull problem and TSP beyond the maximum sizes that they are trained on. Hu et al. (2017); Duan et al. (2019) further apply the Ptr-Net on the bin-packing problem to determine the optimal sequence in which the items are packed into bins. Apart from Ptr-Net, Kool & Welling (2018) solves the TSP based entirely on the attention mechanism similar to the Transformer architecture (Vaswani et al., 2017). The input data is the $(x, y)$ coordinates of all of the nodes, and the model calculates the attention between every pair of nodes. Finally, a sequence of nodes is generated as the optimal result. However, current studies based on Ptr-Net are mainly designed for sequential decisions, which might not fit the pattern for the studied assignment problem. According to Vinyals et al. (2016); Hu et al. (2017), the use of the RNN as an encoder in Ptr-Net makes it difficult to represent long input sequences and suffers heavily from the order of the input data.

The graph-base models, e.g., the graph convolutional network (GCN) and graph attention network (GAT)(Veličković et al., 2018), have also been applied in several studies, because many optimization problems can be represented as a graph. Dai et al. (2017) attempts to solve the minimum vertex cover problem and TSP by embedding the graph with a structure2vec network and iteratively adding the best node to construct the solution using reinforcement learning. Ma et al. (2019) proposes the graph Ptr-Net by using graph embedding as the input of Ptr-Net to solve the TSP. Li et al. (2018) first uses a GCN to generate a large number of candidate solutions to the combinatorial optimization. Then, a guided tree search is performed to refine the solution. However, most mentioned studies address homogeneous graphs without high-dimensional edge features, such as the TSP and VRP problems, where the input is a set of $(x, y)$ coordinates. In the studied problem, the features of orders, items and warehouses are heterogeneous and the relationships between them are important, such as the tiered pricing rules between orders and warehouses, and the inventory between items and warehouses with different sale-forbidden dates. Current studies lack proper consideration of these important high-dimensional edge features and the heterogeneous information between different types of nodes and edges.

## 3 PROBLEM STATEMENT

The mathematical formulation of the studied order fulfillment problem for modern online retailing is formulated in (1a)-(1h). Table 1 lists the notation. For better understanding, a brief introduction on the generalized assignment problem is given in Appendix A.1, which is the basics of the order fulfillment problem.

$$\min_{z_{kl}} \quad \sum_{k \in \mathbb{K}} \sum_{i \in \mathbb{I}} \left( C_{ki}^{(0)} x_{ki} + C_{ki}^{(1)} w_{ki} \right) + \sum_{k \in \mathbb{K}} \sum_{j \in \mathbb{J}} \sum_{t \in \mathbb{T}} P_j \epsilon_{kjt}^+ \tag{1a}$$

$$\text{s.t.} \quad z_{kl} \leq x_{ki}, \qquad \forall k \in \mathbb{K}, \forall i \in \mathbb{I}, \forall l \in \mathbb{L}(i, \cdot), \tag{1b}$$

$$\sum_{k \in K} z_{kl} = 1, \qquad \forall l \in \mathbb{L}, \tag{1c}$$

$$\sum_{t \in \mathbb{T}} d_{klt} = O_l z_{kl}, \qquad \forall k \in \mathbb{K}, l \in \mathbb{L}, \tag{1d}$$

$$\sum_{l \in \mathbb{L}(\cdot, j)} d_{klt} \leq S_{kjt}, \qquad \forall k \in \mathbb{K}, \forall j \in \mathbb{J}, \forall t \in \mathbb{T}, \tag{1e}$$

$$x_{ki} D + w_{ki} \geq \sum_{l \in \mathbb{L}(i, \cdot)} \left( W_j \sum_{t \in \mathbb{T}} d_{klt} \right), \qquad \forall k \in \mathbb{K}, \forall j \in \mathbb{J}, \forall i \in \mathbb{I}, \tag{1f}$$

$$F_{kjt} + \epsilon_{kjt}^+ - \epsilon_{kjt}^- = S_{kjt} - \sum_{l \in \mathbb{L}(\cdot, j)} d_{klt}, \qquad \forall k \in \mathbb{K}, \forall j \in \mathbb{J}, t = 0, \tag{1g}$$

$$F_{kjt} + \epsilon_{kjt}^+ - \epsilon_{kjt}^- = S_{kjt} - \sum_{l \in \mathbb{L}(\cdot, j)} d_{klt} - \epsilon_{kjt-1}^-, \quad \forall k \in \mathbb{K}, \forall j \in \mathbb{J}, t \geq 1 \tag{1h}$$

The objective of the optimization problem (1a) is to minimize the overall fulfillment cost, including the delivery cost and the inventory management cost, by making the optimal assignment between warehouses and suborders. The delivery cost consists of the first weight cost and additional unit cost according to the tiered pricing rule in the logistics. The inventory management cost denotes the loss of the cargo value , which is the result of the items that become sale-forbidden.

Constraint (1b) denotes that if warehouse $k$ serves any suborder in order $i$, then $x_{ki} = 1$. Constraint (1c) denotes that any suborder should be served by one and only one warehouse.

Constraint (1d) denotes that the sum of the selected items before different sale-forbidden periods in warehouse $k$ must equal the gross item quantity of the suborder $l$. Constraint (1e) means that any item $j$ cannot be out-of-stock. Constraint (1f) represents the tiered pricing system in logistics, where the right-hand side of the equation is the total weight of the served part of order $i$ by warehouse $k$. Constraints (1g)-(1h) represent the item picking sequence in the warehouse. Because any item going into sale-forbidden periods can no longer be sold, they must be picked from near to far.

Table 1: Notation List

| PART | DESCRIPTION |
|---|---|
| Sets | $\mathbb{I}, \mathbb{L}, \mathbb{J}, \mathbb{K}, \mathbb{T}$: The set of orders, suborders, items, warehouses and time periods |
| Constants | $D$: The first weight of any package in tiered pricing rules
$O_l$: The item quantity in the suborder $l^1$
$W_j, P_j$: Weight and purchase price of item $j \in \mathbb{J}$
$C_{ki}^{(0)}, C_{ki}^{(1)}$: The first weight cost and additional unit cost of package $(k, i)^2$
$F_{kjt}$: The forecasted sale amount of item $j \in \mathbb{J}$ in warehouse $k \in \mathbb{K}$ in period $t \in \mathbb{T}$
$S_{kjt}$: The inventory of item $j \in \mathbb{J}$ in warehouse $k \in \mathbb{K}$
      that becomes sale-forbidden beyond period $t$ |
| Auxiliary variables | $x_{ki} \in \{0, 1\}$: Whether warehouse $k \in \mathbb{K}$ serves order $i \in \mathbb{I}$, 1: YES, 0: NO
$d_{klt} \in \mathbb{R}^+$: The quantity of items in suborder $l$ served by $k \in \mathbb{K}$
      that becomes sale-forbidden beyond period $t$
$w_{ki} \in \mathbb{R}^+$: The additional weight of package $(k, i)$
$\epsilon_{kjt}^+ \in \mathbb{R}^+$: The quantity of item $j \in \mathbb{J}$ in warehouse $k \in \mathbb{K}$
      that becomes sale-forbidden in period $t \in \mathbb{T}$
$\epsilon_{kjt}^- \in \mathbb{R}^+$: The lack of inventory in $S_{kjt}$ considering picking sequence |
| Decision variables | $z_{kl} \in \{0, 1\}$: Whether warehouse $k \in \mathbb{K}$ serves suborder $l \in \mathbb{L}$, 1: YES, 0: NO |

1 Any suborder $l \in \mathbb{L}$ belongs to a unique order $i \in \mathbb{I}$ and has a group of identical item $j \in \mathbb{J}$.
2 The package $(k, i)$ denotes that part of or all the suborders in order $i \in \mathbb{I}$ are served by the warehouse $k \in \mathbb{K}$ and are delivered as one package.

Note that $z_{kl}$ is the key decision variable, and it represents the assignment between warehouses and suborders. When we make decision, two constraints are the most important: (1) Each suborder should be assigned to only one warehouse, represented by (1c); (2) The item inventory in the assigned warehouse must not be less than the demand of the suborder, represented by (1d)-(1e). Other variables and constraints are auxiliary. For example, $d_{klt}, \epsilon_{kjt}^+, \epsilon_{kjt}^-$ are used mainly to represent the item picking sequence; and $x_{ki}, w_{ki}$ are used to represent the difference between the first weight and the additional weight in the tiered pricing rules.

## 4 SOLUTION FRAMEWORK

In the proposed solution, we first map this problem into a tripartite heterogeneous graph as the input to the graph ML model, which has the advantage of being permutation-invariant to the input order. Then, we design the edge-feature-embedded graph attention to learn the hidden state of the graph. Last, we use the attention between the hidden state of the orders and warehouses to make the assignment, which can fit cases with varying sizes of orders, items and warehouses.

During the offline training, the exact B&B method is applied to solve the problem instances to generate the optimal assignment results as the training labels. In the stage of online inference, we directly predict the suborder-warehouse assignment through the graph ML model. Because the proposed graph ML model is size-invariant, it can generalize to cases of larger sizes that have never been seen during training.

### 4.1 TRIPARTITE GRAPH REPRESENTATION

First, we novelly represent the order fulfillment problem with a tripartite graph in Figure 1a, where vectors $\boldsymbol{h}, \boldsymbol{e}$ represent the features of nodes and edges, respectively. Compared to the sequential representation, the graph representation has the advantage of being permutation-invariant, which means that different input data orders correspond to the same graph. The three disjoint sets in the graph are the orders (decomposed into suborders), items and warehouses.

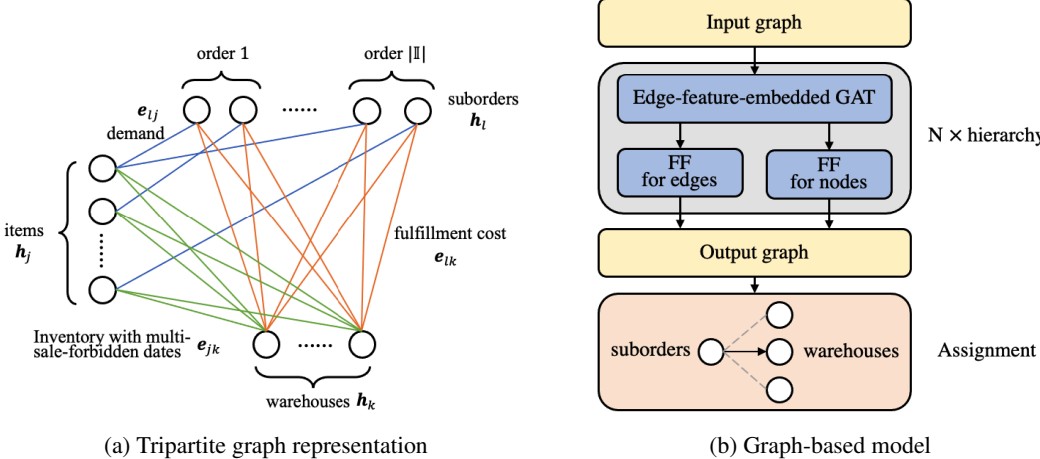

(a) Tripartite graph representation            (b) Graph-based model

Figure 1: The tripartite graph representation and the graph-based machine learning model

Different sets of nodes have different features. The feature $\boldsymbol{h}_l \in \mathbb{R}^N, l \in \mathbb{L}$ of suborder node $l$ is an $N$-dimensional one-hot vector, representing the order that it belongs to. The suborders under the same order share the same feature. $N$ denotes the maximum size of orders the model is designed to handle. The feature $\boldsymbol{h}_k \in \mathbb{R}^M, k \in \mathbb{K}$ of warehouse node $k$ is an $M$-dimensional one-hot vector, representing the different warehouse identification codes (ids). $M$ denotes the maximum size of the warehouses that the model is designed to handle. The feature $\boldsymbol{h}_j = [P_j, W_j], j \in \mathbb{J}$ of item node $j$ is a two-dimensional vector that consists of the purchase price and weight of the corresponding item.

The edges between different set of nodes also represent different information. The feature $\boldsymbol{e}_{kl} = [C_{ki}^{(0)}, C_{ki}^{(1)}], k \in \mathbb{K}, l \in \mathbb{L}(i, \cdot)$ between suborder $l$ and warehouse $k$ represents the tiered delivery cost. Note that the first weight and additional weight cost from one warehouse to the suborders under the same order are identical. The feature $\boldsymbol{e}_{lj} = [O_l], j \in \mathbb{J}, l \in \mathbb{L}(\cdot, j)$ of the edges between suborder $l$ and item $j$ represents the quantity of items that each suborder contains. The feature $\boldsymbol{e}_{kj} = [S_{kj0}, F_{kj0}, ..., S_{kjt}, F_{kjt}, ...], k \in \mathbb{K}, j \in \mathbb{J}$ of edges between warehouse $k$ and item $j$ is a vector with the length of $2 \times |\mathbb{T}|$, which represents the inventory and forecasted saleplan in different sale-forbidden dates.

## 4.2 THE GRAPH-BASED MACHINE LEARNING MODEL

To learn the hidden states from the original tripartite graph and to make an assignment, we propose a hierarchical graph-based machine learning model in Figure 1b. The model consists of three important components. The first component is the edge-feature-embedded GAT layer, which follows the work of Veličković et al. (2018) but uses a novel attention mechanism to learn the high-dimensional edge features and the heterogeneous information in the graph. Next, a feed forward (FF) layer makes a nonlinear transformation of the node and edge features. Last, an "assignment" layer generates the assignment between suborders and warehouses by computing the attention coefficients of each pair. It is worthwhile to note that the attention mechanism is used throughout the model and no transformation depends on the size of the graph, which makes the model able to address varying sizes of orders, items and warehouses.

### 4.2.1 EDGE-FEATURE-EMBEDDED GAT LAYER

The input to the edge-feature-embedded GAT is a set of node feature vectors $\{\boldsymbol{h}_l, \boldsymbol{h}_j, \boldsymbol{h}_k\}$ and edge feature vectors $\{\boldsymbol{e}_{lj}, \boldsymbol{e}_{lk}, \boldsymbol{e}_{kj}\}, \forall l \in \mathbb{L}, j \in \mathbb{J}, k \in \mathbb{K}$. Here, we will use the transformation of nodes that represent suborders $\boldsymbol{h}_l, \forall l \in \mathbb{L}$ to illustrate how we properly embed the edge features.

First, we embed the edge features in the computation of the attention coefficients. For any node $l \in \mathbb{L}$, we compute the attention coefficients between $\boldsymbol{h}_l$ and $\boldsymbol{h}_j, \boldsymbol{h}_k$ as follows:

$$c_{lj} = \text{score}(\boldsymbol{h}_l, \boldsymbol{h}_j, \boldsymbol{e}_{lj}) = \sigma\left(\boldsymbol{V}_{\mathbb{JL}}^{\top}(\boldsymbol{h}_l\|\boldsymbol{h}_j\|\boldsymbol{e}_{lj})\right), \forall j \in \mathbb{J} \tag{2}$$

$$c_{lk} = \text{score}(\boldsymbol{h}_l, \boldsymbol{h}_k, \boldsymbol{e}_{lk}) = \sigma\left(\boldsymbol{V}_{\mathbb{KL}}^{\top}(\boldsymbol{h}_l\|\boldsymbol{h}_k\|\boldsymbol{e}_{lk})\right), \forall k \in \mathbb{K} \tag{3}$$

Here, the attention mechanism is a single feed-forward layer with the nonlinear activation function $\sigma$. $\boldsymbol{V}_{\mathbb{JL}}$ and $\boldsymbol{V}_{\mathbb{KL}}$ denote the weight vectors with respect to heterogenous node pairs. For any $j \in \mathbb{J}$ and $k \in \mathbb{K}$, the weight vectors remain the same and thus are invariant to the sizes of the node sets. $\|$ represents the concatenation operation. The attention coefficients indicate the importance between any two nodes considering the edge feature that links them and are computed only if the nodes are connected in the graph. The coefficients are then normalized across the same type of neighborhood nodes using the softmax function:

$$a_{lj} = \text{softmax}_j(c_{lj}) = \frac{\exp(c_{lj})}{\sum_{j\in J}\exp(c_{lj})}, \forall j \in \mathbb{J} \tag{4}$$

$$a_{lk} = \text{softmax}_k(c_{lk}) = \frac{\exp(c_{lk})}{\sum_{k\in K}\exp(c_{lk})}, \forall k \in \mathbb{K} \tag{5}$$

Second, we embed the edge features into nodes using a linear combination by the attention coefficients to update $\boldsymbol{h}_l, \forall l \in \mathbb{L}$:

$$\widehat{\boldsymbol{h}}_l = \boldsymbol{h}_l\left\|\left[\sum_{k\in\mathbb{K}} a_{lk}(\boldsymbol{h}_k\|\boldsymbol{e}_{lk})\right]\right\|\left[\sum_{j\in\mathbb{J}} a_{lj}(\boldsymbol{h}_j\|\boldsymbol{e}_{lj})\right] \tag{6}$$

Two things should be noted in (6). First, the updated node feature is concatenated with three parts. The first part is the self-feature, while the others are the combination of the feature of items and warehouses. The reason is that the graph we address contains heterogeneous information and should be treated appropriately. Second, the edge feature is again concatenated with the node feature apart from that in (2)-(3). One special case can illustrate the necessity behind this step. Consider a node that has only one neighborhood. The attention coefficients will always be 1.0 for this neighborhood, regardless what the edge feature is. If we do not concatenate the edge feature according to the method of (6), we lose the information on the edge between them. For other cases, although (2)-(3) have already embedded the edge features, the concatenation of edge features in (6) can further strengthen the explicit representation of the edge information.

By applying the transformation method (2)-(6) on nodes $\boldsymbol{h}_j, \forall j \in \mathbb{J}$ and $\boldsymbol{h}_k, \forall k \in \mathbb{K}$, we obtain the updated node feature $\widehat{\boldsymbol{h}}_j$ and $\widehat{\boldsymbol{h}}_k$, which, in combination with $\widehat{\boldsymbol{h}}_l$, forms the output of the layer.

### 4.2.2 FEED-FORWARD LAYER FOR NODES AND EDGES

Note that the edge-feature-embedded GAT layer only makes a linear combination of features in equation (6). Therefore, we further use a feed-forward layer to transform the features of nodes and edges with nonlinearity. We still take the nodes that represent suborders as an example, of which the updated feature after the GAT layer is $\widehat{\boldsymbol{h}}_l$. The transformation through the feed-forward layer is as follows:

$$\widetilde{\boldsymbol{h}}_l = \sigma\left(\boldsymbol{W}_{\mathbb{L}}\widehat{\boldsymbol{h}}_l\right), \forall l \in \mathbb{L} \tag{7}$$

For the edges, the GAT layer does not make a transformation and thus we have the following transformation for the edges that connect any suborder $l \in \mathbb{L}$:

$$\widetilde{\boldsymbol{e}}_{lj} = \sigma\left(\boldsymbol{W}_{\mathbb{JL}}\boldsymbol{e}_{lj}\right), \forall j \in \mathbb{J} \tag{8}$$

$$\widetilde{\boldsymbol{e}}_{lk} = \sigma\left(\boldsymbol{W}_{\mathbb{KL}}\boldsymbol{e}_{lk}\right), \forall k \in \mathbb{K} \tag{9}$$

where $\boldsymbol{W}_{\mathbb{JL}}, \boldsymbol{W}_{\mathbb{KL}}$ are the transformation matrices for the edges between suborders and items, suborders and warehouses. Note that for any suborder node and the connected edges, the weight matrix $\boldsymbol{W}_{\mathbb{L}}, \boldsymbol{W}_{\mathbb{JL}}, \boldsymbol{W}_{\mathbb{KL}}$ are the same and are thus invariant to the size of node set.

The outputs of FFN are $\{\widetilde{\boldsymbol{h}}_l, \widetilde{\boldsymbol{h}}_j, \widetilde{\boldsymbol{h}}_k\}$ and $\{\widetilde{\boldsymbol{e}}_{lj}, \widetilde{\boldsymbol{e}}_{lk}, \widetilde{\boldsymbol{e}}_{kj}\}, \forall l \in \mathbb{L}, j \in \mathbb{J}, k \in \mathbb{K}$. They formulate a new graph with the same topology as the input graph and will be used as the input for the next hierarchy for further transformation.

### 4.2.3 ASSIGNMENT LAYER

To assign suborders to warehouses in size-variant cases, we again use the attention mechanism as a "pointer" Vinyals et al. (2015) to point each suborder to the appropriate warehouse. First, we calculate the attention coefficients on the output graph between each pair of suborder $l$ and warehouse $k$ using the dot product:

$$c_{lk} = \text{score}\,(\boldsymbol{h}_l, \boldsymbol{h}_k) = \boldsymbol{h}_l^\top \boldsymbol{h}_k, \forall l \in \mathbb{L}, k \in \mathbb{K} \tag{10}$$

Next, we apply the softmax function to normalize over all of the warehouses and, for any suborder node $l \in \mathbb{L}$, we use the $p_{lk}$ as a distribution over the warehouses.

$$p_{lk} = \text{softmax}_k(c_{lk}) = \frac{\exp(c_{lk})}{\sum_{k \in \mathbb{K}} \exp(c_{lk})} \tag{11}$$

The largest probability $\arg\max_{k \in \mathbb{K}} p_{lk}$ is the selected suborder-warehouse assignment by the ML model. Note that in this step, there is no weight matrix to be trained and thus, the approach is invariant to the sizes of the suborders and warehouses.

### 4.3 OFFLINE MODEL TRAINING

In the stage of offline model training, we use supervised learning. The cross-entropy is applied as the loss function. The total loss in an instance should account for all of the suborders as follows:

$$\text{loss} = -\sum_{l \in \mathbb{L}} \sum_{k \in \mathbb{K}} y_{lk} \log(p_{lk}) \tag{12}$$

where $y_{lk}$ is the real label that denotes whether suborder $l$ is assigned to warehouse $k$. Real labels are generated by using the exact B&B method to solve the MIP (1a)-(1j) under different input instances.

### 4.4 ONLINE MODEL INFERENCE

In the stage of online inference, each input case is formulated into the tripartite graph, and the offline-trained graph-based model is directly used to compute the attention coefficients between each pair of suborders and warehouses. To guarantee the feasibility in the online order fulfillment decision, the most important constraints (1c)-(1e) must be satisfied. We use the following masking procedure in finding the largest probability $\arg\max_{k \in \mathbb{K}} p_{lk}$: (1) Select the suborder $l = \arg\max_l O_l$; (2) The warehouse node $k \in \mathbb{K}$ with insufficient inventory ($O_l > \sum_{t \in \mathbb{T}} S_{kjt}$) is not allowed to be visited; (3) Find the best warehouse $k = \arg\max_{k \in \mathbb{K}} p_{lk}$ and reduce the inventory of item $j$ from the nearest sale-forbidden date until the quantity equals $O_l$.

## 5 EXPERIMENTS

### 5.1 DATASETS, BASELINES AND METRICS

We generate one training and three testing datasets with varying scales. The ML model is trained only on the training set and then tested on the three test sets. The training dataset contains 100,000 instances of the order fulfillment problem with the optimal assignment results as labels, which are obtained by solving the MIP formulation (1a)-(1h) using the open-source SCIP optimization suite Gleixner et al. (2017). The problem size in the test cases is larger than that in the training dataset.

We report four baselines in the experiments. The first is the exact branch-and-bound (B&B) method. The second is the heuristic method that is currently applied in practice (The detailed algorithm is described in appendix A.2.) The third baseline is to use the Ptr-Net based on supervised learning, which is also our first attempt facing the order fulfillment problem. To compare the effect of considering the high-dimensional edge features and heterogeneous information in our model, we add the fourth baseline by replacing the edge-feature-embedded graph encoders with the original GAT network Veličković et al. (2018) in our solution framework.

The evaluation metrices are the computation time and the gap in the total cost between the model's result and the optimal solution (obtained by SCIP with a time limit of 15 minutes). The total cost

Table 2: Description of datasets and experimental results

| DATASET | DESCRIPTION | METHOD | COST GAP | INFERENCE TIME(ms) |
|---------|-------------|--------|----------|---------------------|
| TARIN | Size: 100,000 $\|\mathbb{I}\| : 6 \sim 20$ $\|\mathbb{J}\| : 30 \sim 40$ $\|\mathbb{K}\| : 4 \sim 8$ | — | — | — |
| TEST 1 | Size: 10,000 $\|\mathbb{I}\| : 6 \sim 20$ $\|\mathbb{J}\| : 30 \sim 40$ $\|\mathbb{K}\| : 4 \sim 8$ | Exact B&B Heuristic Ptr-Net GAT **Our method** | 0% 12.6% 15.6% 16.4% **2.1%** | 450 1.8 31 2.5 **2.7** |
| TEST 2 | Size: 10,000 $\|\mathbb{I}\| : 20 \sim 50$ $\|\mathbb{J}\| : 70 \sim 90$ $\|\mathbb{K}\| : 8 \sim 12$ | Exact B&B heuristic Ptr-Net GAT **Our method** | 0% 11.9% 14.8% 16.1% **2.5%** | 3,200 2.7 78 4.9 **5.1** |
| TEST 3 | Size: 100 $\|\mathbb{I}\| : 50 \sim 100$ $\|\mathbb{J}\| : 100 \sim 120$ $\|\mathbb{K}\| : 12 \sim 20$ | Exact B&B heuristic Ptr-Net GAT **Our method** | 0% 11.4% 13.8 15.1% **3.0%** | 62,000 5.8 210 12 **13** |

(a) Results on test set 1     (b) Results on test set 2     (c) Results on test set 3

Figure 2: Boxplot of inference results on test sets

is computed using equation (1a) based on the order fulfillment result. A smaller gap means a better assignment policy.

$$\text{gap} = \frac{\text{cost}_{\text{model}} - \text{cost}_{\text{MIP}}}{\text{cost}_{\text{MIP}}} \tag{13}$$

## 5.2 ARCHITECTURE AND HYPERPARAMETERS

In the experiments, we apply $N = 3$ graph encoders, without using an extensive hyperparameter search and the multi-head mechanism. The nonlinear activation function in both the edge-feature-embedded GAT layer and the FFN layer is the exponential linear unit (ELU) Clevert et al. (2015). During the training, we apply a dropout ratio of 0.1 to the FFN inputs. The model is trained with the Adam optimizer with a learning rate of 0.003 and a batch size of 256.

## 5.3 RESULTS

The details of the datasets and the results of the experiments are summarized in Table 2 . Figure 2 compares the performance of all of the methods on the test datasets. Figure 3 gives an example of the model-predicted assignment result from the 3rd test case.

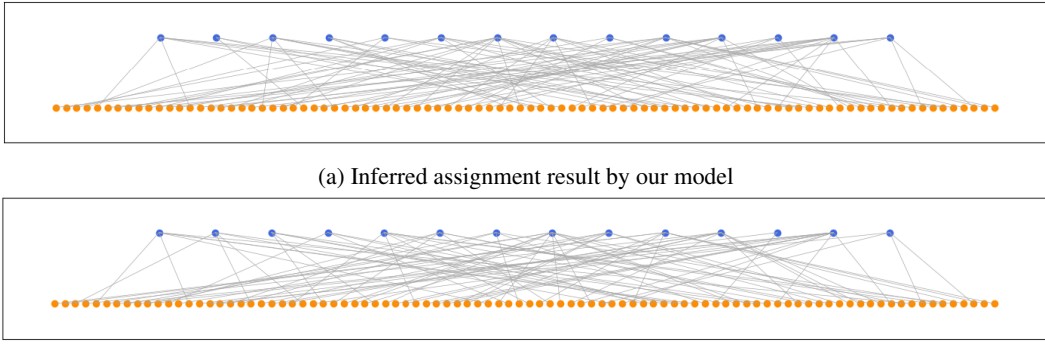

(a) Inferred assignment result by our model

(b) Optimal assignment result by solving MIP

Figure 3: An example case with 60 orders (92 suborders, orange points) and 14 warehouses (blue points). The cost gap in this example is 3.52%.

From the results, we note that our method achieves the smallest cost gap and substantially outperforms the heuristic method, Ptr-Net and the original GAT without embedded edge features. In Figure 2, we find that our method also has the smallest variance at the stage of inference, which means better robustness. In test sets 2 and 3, where the sizes of the problem instances are completely unseen during training, our method still has the best performance in extrapolating.

The inference time of these methods is also compared. We note that, although the heuristic method has the lowest computational cost because of its simple rules, our method is able to make inference in several milliseconds. Compared to directly solving MIP, our method achieves tens of thousands of times of acceleration in the computation time and meets the criteria for real-time applications. The reason behind this outcome is that the graph encoders compute the assignment results through matrix multiplications. In contrast, the exact B&B method iteratively searches the results in the feasible region and is thus much slower than our method.

The reason for the bad performance with the original GAT network is obvious, because the important high-dimensional edge features are ignored in the model. However, it is worthwhile to note that Ptr-Net does not perform well in the studied task. We think that three reasons might lead to this result. First, Ptr-Net uses an RNN to encode the input data, which is heavily influenced by the sequence order and length. For example, in the second test case, the length of the input sequence is approximately 800, which could be too long for the RNN to learn the hidden states. Second, the assignment problem is not a sequential decision problem that differs from TSP and VRP, which is incompatible with the structure of Ptr-Net. Third, the output of Ptr-Net is generated one-by-one, depending on the previous output sequence, which leads to a larger inference time.

## 6   CONCLUSIONS

In this paper, we propose a machine learning framework to solve the order fulfillment problem—a special type of combinatorial optimization—for real-time decisions in online retailing. Experiments show that our method substantially outperforms the baseline heuristic method and the cost gap between MIP optimal solution is less than 5%. Our method also realizes thousands of times of acceleration compared with directly solving the MIP, which meets the criteria for real-time online decision in milliseconds. To meet the characteristics of the studied problem, the edge-feature-embedded graph encoder is developed, which takes the high-dimensional edge features and heterogeneous information into the graph attention mechanism and can address problem instances with varying input lengths and output dictionary sizes.

Apart from solving the studied order fulfillment problem, we envision that the edge-feature-embedded heterogenous graph attention can be further applied in other combinatorial optimization problems such as the TSP, VRP and the generalized assignment problem, where our method can include consideration of the high-dimensional nonlinear route cost or other complex scenarios.

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

## A  APPENDIX

### A.1  GENERALIZED ASSIGNMENT PROBLEM

The simplest form is called the assignment problem, in which the knapsack for items is uncapacitated. The assignment problem is a special case of the minimum cost flow problem and can be formulated as a linear programming problem without using integers. Several exact and fast algorithms have been developed such as the Hungarian method, the auction algorithm. Compared to the assignment problem, the generalized assignment problem has capacitated knapsacks and other real-world constraints, which make it vary difficult to solve.

The generalized assignment problem describes the assignments between items and bins, where the bins have a capacity limit. There could exist other complex constraints for the items and bins. The problem can be formulated as a mixed integer program:

$$\min_{z_{ij}} \quad \sum_{i=1}^{m}\sum_{j=1}^{n} c_{ij} z_{ij} \tag{14a}$$

$$\text{s.t.} \quad \sum_{j=1}^{n} w_{ij} z_{ij} \le t_i, \quad i = 1, ..., m, \tag{14b}$$

$$\sum_{i=1}^{m} z_{ij} = 1, \qquad j = 1, ..., n, \tag{14c}$$

$$f(z_{ij}) \leq 0, \qquad i = 1, ..., m, j = 1, ..., n, \qquad (14d)$$

$$g(z_{ij}) = 0, \qquad i = 1, ..., m, j = 1, ..., n, \qquad (14e)$$

$$z_{ij} \in \{0, 1\} \qquad i = 1, ..., m, j = 1, ..., n \qquad (14f)$$

In the above formulation, we have $n$ items and $m$ bins. For bin $i$, each item has a cost $c_{ij}$ and a weight $w_{ij}$. A solution is an assignment from items to bins. The objective is to minimize the cost of the assignment. The constraint (14b) denotes that any bin $i$ has a capacity $t_i$. The constraint (14c) denotes that any item can only be assigned to one bin. The equation (14b)-(14c) are the fundamental constraints in the generalized assignment problem, while (14d)-(14e) represent other general constraints for the problem.

The explorations on the exact method to solve the generalized assignment problem are mainly based on decomposition techniques. Wu et al. (2018) applies Benders decomposition and Lagrangian relaxation to reduce the scale of the MIP problem. Ghoniem et al. (2016) considers the total space allocation constraints and modifies the branch-and-price method. According to the paper, it costs half an hour to reach a nearly-optimal solution with only 15 items and 25 knapsacks.

Several heuristics are developed for the generalized assignment problem as well. Sharkey & Romeijn (2010); Cohen et al. (2006) use greedy methods to translate the algorithm for the knapsack problem into an approximation method for a generalized assignment problem. Fleischer et al. (2006) further proves the performance of the proposed heuristics by justifying an optimal gap less than $1/e$. Sethanan & Pitakaso (2016); Özbakir et al. (2010) use differential evolution and the bees algorithm respectively, with the help of local search techniques. An average optimality gap of 12.5% is achieved. The downside of heuristics is that the approach requires an enormous amount of human knowledge and might not reach the state-of-art performance.

## A.2 Heuristic method for the order fulfillment problem

The algorithm below is the current heuristic method for the order fulfillment problem. The remaining inventory near sale-forbidden dates among different warehouses is the first priority, because the cargo value loss is usually larger than the delivery fee. Next, the delivery fee for each warehouse is compared, while considering the tiered pricing rule, and the relationships between the orders and suborders. Finally, the corresponding item inventory is reducted from the best selected warehouse.

---

**Algorithm 1:** Heuristic method for the order fulfillment problem

---

**Result:** $z_{kl}$
$\mathbb{I}' = \mathbb{I}$
**while** $\mathbb{I}' \neq \varnothing$ **do**
    $\mathbb{K}' = \varnothing$
    $w_k = 0, \forall k \in \mathbb{K}$
    $i = \arg\max_i \sum_{l \in \mathbb{L}(i,\cdot)} O_l$
    **for** $l \in \mathbb{L}(i, \cdot)$ **do**
        // Select the best warehouse
        **if** $\exists t \in \mathbb{T}(t \neq |\mathbb{T}| - 1), s.t. S_{kjt} > 0$ **then**
            $k^* = \arg\max_k \sum_{t \in \mathbb{T}} S_{kjt}$
        **else**
            **for** $k \in \mathbb{K}, s.t. \sum_{t \in \mathbb{T}} S_{kjt} > O_l$ **do**
                **if** $k \in \mathbb{K}'$ **then**
                    **if** $w_k + W_j O_l \leq D$ **then**
                        $c_{kl} = 0$
                    **else**
                        $c_{kl} = C_{ki}^{(1)}(w_k + W_j O_l - D)$
                    **end**
                **else**
                    **if** $W_j O_l \leq D$ **then**
                        $c_{kl} = C_{ki}^{(0)}$
                    **else**
                        $c_{kl} = C_{ki}^{(0)} + C_{ki}^{(1)}(W_j O_l - D)$
                    **end**
                **end**
            **end**
            $k^* = \arg\min_k c_{kl}$
            $w_{k^*} = w_{k^*} + W_j O_l$
            $\mathbb{K}' = \mathbb{K}' \cup k^*$
        **end**
        $z_{k^* l} = 1$
        // Reduct inventory
        $r_l = O_l$
        **for** $t \in \mathbb{T}$ **do**
            **if** $0 \leq r_l < S_{kjt}$ **then**
                $S_{kjt} = S_{kjt} - r_l; r_l = 0$
            **else**
                $r_l = r_l - S_{kjt}; S_{kjt} = 0$
            **end**
         **end**
    **end**
    $\mathbb{I}' = \mathbb{I}' \setminus i$
**end**

---

