# OpenReview forum: "Learning to Solve an Order Fulfillment Problem in Milliseconds with Edge-Feature-Embedded Graph Attention"
_ICLR.cc/2022/Conference — ICLR 2022 Submitted_

### Official Review · Reviewer_pGWk · 2021-11-01

**Correctness:** 3
**Technical Novelty And Significance:** 3
**Empirical Novelty And Significance:** 3
**Recommendation:** 6
**Confidence:** 3

**Main Review:**

Strength:

- The model seems novel and is interesting.

- The problem is well motivated: using efficient machine learning models to solve the originally NP-hard combinatorial order fulfillment problem.

- The reported performance seems promising. The cost gap is only 2 - 3% compared to the optimal solution, and the inference time is minimal, similar to previous heuristic approaches.

Weakness:

- The authors should describe how they generated the 100,000 training dataset instances. I could not find the generative process. Please let me know if I missed it. Is there any unmentioned pattern in the training dataset, or the problem instances can be arbitrary? Please try to discuss any potential limitation.

- In addition, if as the authors implied, the model performs well even on unseen problem instances (test 2 and 3), they should consider release the algorithm for generating the synthetic training datasets, because then people can simply train the same model and apply it to their own order fulfillment problems.

- Table 2: is the listed inference time for every instance problem in the test set?

- Have the authors tried their model on some real world datasets?

**Summary Of The Paper:**

This paper proposes a heuristic graph based machine learning model, which can be used to solve the class of order fulfillment problems. The original mixed integer programming problem is known to be NP-hard, and thus computationally intractable. The authors treat each instance of the order fulfillment problems as a tripartite graph, which serves as the input of the proposed model.

The proposed model consists of three parts: a graph attention network (GAT) to embed the features, a feed forward (FF) layer, and an assignment layer generating the output.

**Summary Of The Review:**

I think the idea of converting an order fulfillment problem to a tripartite graph, which is then fed to a GAT, is interesting. However it is hard to know if the model could generalize well to other synthetic datasets or real world problems, since the authors only used one synthetic training dataset in their experiments, and the generative process is unknown.

---

### Official Review · Reviewer_ciU3 · 2021-11-02

**Correctness:** 3
**Technical Novelty And Significance:** 2
**Empirical Novelty And Significance:** 3
**Recommendation:** 3
**Confidence:** 3

**Main Review:**

**Strengths**
- The paper presents a new approach (inspired by previous work) using ML for the order fulfillment problem, which I can imagine appears in practice in online retail
- The paper makes suitable and mostly principled modifications to existing architectures to be suitable for the order fulfilment problem, especially integrating edge features and making the model suitable for a combination of different node types (orders, suborders and warehouse).
- The problem is clearly and formally specified through a mathematical formulation
- The model is clearly explained on the right level of abstraction for the reader familiar with graph neural networks
- Experiments illustrate potentially good results, achieving good quality solutions in time orders of magnitude faster than MIP solving, and with better quality than a heuristic. It would be interesting if the paper could explore why the quality of the solution is much better than the heuristic, which seems a reasonable strategy.

**Weaknesses**
- The positioning of the paper is somewhat unclear to me: does this paper present an existing problem or introduce/formalize a new problem? Does it address a real application (as suggested by 'heuristic used in practice', in which case I'd like to see more details on where/how it is used in practice, to judge the relevance) or is it fictional and mainly about the methodology (this is the impression I get from the related works section)?
- The motivation is a bit weak in my opinion: why do we require solving in 'tens of milliseconds', i.e. 'realtime', whereas (I assume) fulfillment takes at least a few hours? Even with 'hundreds of orders' per seconds, we could solve many instances in parallel and get 2-3% better solutions (= 2-3% lower cost which I think is a lot in practice!) by using SCIP for 60 seconds instead of the model.
- The empirical evaluation is incomplete: SCIP gives better results whereas the model is faster, so it is unclear what is preferred. Both methods should *at least* be compared using the same computation time (e.g. SCIP may find solutions with 2-3% gap early as well), and ideally a full time vs performance curve should be reported.
- Whereas a graph network, in principle, is permutation invariant and can scale to any size, the model still depends on the permutation of orders for the one-hot feature indicating the order, which also limits the maximum number of orders the model can handle.
- Some claims are unsupported or incorrect, e.g. "The order fulfillment problem is one of the fundamental optimization problems in supply chain management" could at least use a reference, "tens of thousands times faster" does not accurately reflect the results (200-5000x).
- Some parts are unclear/not explained or introduced, e.g. tiered pricing rules and some relevant details are unclear/missing, e.g. how are the parameters for the 100.000 training instances (and the test instances) generated?
- Overall, the paper presents a relatively straightforward application of an existing idea (graph neural network with supervised learning). Whereas the problem requires suitable modifications to the standard GNN/GAT, this is not the first paper to handle node types and edge features (e.g. first hit on Google gives [1]) and the method is relatively straightforward.

**Minor comments/questions**
- I think the paper means 'neighbor' in some cases where 'neighborhood' is written.
- TARIN -> TRAIN (Table 2)
- Why is a SCIP time limit of 15 minutes needed since according to Table 2 instances of that size are solved in 0.45 seconds on average?
- If a time limit is used and has effect, I suggest not to call SCIP results optimal

[1] Frederik Diehl, Applying Graph Neural Networks on Heterogeneous Nodes and Edge Features, NeurIPS 2019 Graph Representation Learning Workshop

**Summary Of The Paper:**

This paper presents a graph based deep learning model for solving the order fulfillment problem of assigning retail (sub-)orders to different warehouses for fulfilment. It formulates the problem as a Mixed Integer Problem (MIP, although I think it is fully integer) which can be solved using a (open-source) MIP solver such as SCIP. Following up on related work, the authors train a graph neural network, specifically designed to incorporate edge features and different types of nodes, to predict the optimal solution, using supervised learning on a dataset of 100.000 generated instances solved with SCIP. Empirically, on three different test problem sizes, the trained model provides solutions within 2-3% to the results found by SCIP, while being a 2-4 orders of magnitude faster.

**Summary Of The Review:**

Overall, this is a nice paper that applies an interesting learning based approach (inspired by previous works) to the order fulfillment problem, with encouraging results. However, I think the paper needs to improve in multiple aspects to address the mentioned weaknesses: positioning, motivation and empirical evaluation. Even then, I am not sure if ICLR is the most suitable venue as I consider this a relatively straightforward application to the order fulfillment problem, admitting that it does require suitable modifications to the graph neural network architecture (but nothing groundbreaking in my opinion). Whereas ICLR welcomes applications as well, I think in that case it should consider a problem which is more established and thus has stronger baselines from other papers, such that it really convinces on the need/benefits of using ML.

---

### Official Review · Reviewer_PyJm · 2021-11-03

**Correctness:** 3
**Technical Novelty And Significance:** 2
**Empirical Novelty And Significance:** 2
**Recommendation:** 5
**Confidence:** 4

**Main Review:**

**Strengths**

- Motivation: a very interesting and challenging problem that is worth considering for ML methods.
- Clarity: overall, the paper is well-written and easy to read.
- Experiments: the proposed GNN works well on instances larger than those seen in training and compares favorably to the baselines.

**Weaknesses**

1. Knowing max number of orders (section 4.1) goes a bit against your claim about size-invariance of the proposed model.

2. Datasets: what is the instance generation procedure? Has it been used to generate hard instances of the order fulfillment problem in the literature before? Without an answer to both of these questions, it is difficult to put your results in context.

3. Baseline heuristic: In appendix A.2, you say that Algorithm 1 “is the current heuristic method for the order fulfillment problem”; can you cite a source for this statement? Where does this heuristic come from?

4. Inference time for Exact B&B in Table 2: is the time reported for this method the total time it requires to prove optimality? If so, then the inference time reported for exact B&B is unfair. You should instead track the incumbent solutions it finds over time and measure the cost gap at different time limits, e.g., the cost gap of exact B&B if it is given 10ms only, then 20ms only, etc. It is often the case that MIP solvers will find a near-optimal feasible solutions early on, then use most of the remaining time to prove optimality. However, your paper focuses on the former aspect not the latter, hence my suggestion here.

5. Sample complexity: you use 100k training instances but no results are reported for smaller datasets. How much data does your GNN need to achieve good performance? This is particularly important because you require optimal solutions as labels!

**Other questions/comments**

6. Section 4.4. Does the “masking procedure” always work, i.e., is a feasible solution guaranteed?

7. Figure 3 is not very useful as there are many edges and it is impossible to see how the two solutions differ. I recommend either removing this figure or finding a better visualization of the differences between solutions.

**Typos**

- Eq. (1c): I think the subscript in the sum should have the same K as the one in notation Table 1 (1st row).
- Table 2: “TARIN” —> “TRAIN”

**Summary Of The Paper:**

A Graph Neural Network (GNN) model is designed to enable the supervised learning of optimal solutions for an order fulfillment problem in supply chain management. This is a problem that must be solved in real-time, making a GNN whose forward computations are quick an attractive option.

The proposed GNN model has three types of nodes corresponding to orders, items, and warehouses. Edges between the nodes of different types represent relationships between them (e.g., an item is in an order). Nodes and edges may have feature vectors representing additional information, e.g., the identity of a warehouse or certain costs on the edges. The GNN is designed to take these features into account.

The model is trained using optimal solutions as labels in a binary classification formulation. Predictions are post-processed to guarantee a feasible solution in the constraints of the order fulfillment problem.

Experimental results on randomly generated instances show that a GNN trained on small instances of this problem performs well even on larger instances, yielding near-optimal solutions in a short amount of time compared to simple GNN solutions, a non-ML heuristic, and an exact solver.

**Summary Of The Review:**

There is an challenging and important problem here and a promising method. However, the experiments are rather limited in scope. The modeling contribution is interesting but it is unclear if it generalizes to other combinatorial optimization problems.

---

### Official Review · Reviewer_qjBL · 2021-11-05

**Correctness:** 3
**Technical Novelty And Significance:** 3
**Empirical Novelty And Significance:** 3
**Recommendation:** 3
**Confidence:** 4

**Main Review:**

The first strength of the paper is that it is built upon best practices found in last 3 years of research on ML-based Combinatorial Optimization: imitation learning when a good expert is available, GNN for feature extraction, and GAT for the architecture choice. Authors should've referenced Gasse et al https://arxiv.org/abs/1906.01629 and Nair et al https://arxiv.org/abs/2012.13349 which use the similar strategy for the more generic MIP problem, however.

The second strength of the paper is the practicality of the problem addressed. This version of the fulfillment problem has nontrivial components such as order decomposition to sub-orders and sale-forbidden dates. Therefore, I expect the proposed method to generalize well for practical problems in the real world. There aren't enough discussions on why this particular version of the problem was chosen, however. I encourage authors to include more discussion on what versions of fulfillment problems are popular in which applications, and how representative the given problem formulation is.

One notable weakness of this paper is the experimental setup. First, some of the important details are missing. Authors only mention that training dataset consists of 100k instances and there are three test cases, but don't really mention anything about from where the data & problem came, or what their characteristics are, other than the problem size itself. Second, it is unclear how strong baselines are. SCIP has a variety of parameters to optimize- authors don't mention which parameters they used. Depending on parameters of the heuristic, SCIP would make a different tradeoff between accuracy & efficiency, and therefore it would be more informative if SCIP's performance at variety of levels of efficiency could be reported. Third, since exact B&B is the expert policy, "learning to branch" type of algorithms like Gasse et al https://arxiv.org/abs/1906.01629 and Nair et al https://arxiv.org/abs/2012.13349 which learn directly from each branching decision of the expert are critical baselines to have. Fourth, the heuristic baseline needs more explanation on what heuristics are available on the literature, and why this particular version of the heuristic was chosen. Without such discussion, we cannot really learn about the comparison with the heuristic. Authors also mention the heuristic is "currently applied in practice"; this hints authors might be describing a production system. In that case, more description of the system will be needed.

Some of the technical claims are questionable. Authors mention that the model requires no transformation that depends on the size of the graph, which allows the model to generalize to varying sizes of orders, items, and warehouses (Section 4.2). But since that suborder node features are one-hot-vectors across the fixed number of nodes $N$, and warehouses are represented as one-hot over $M$, it is unclear to me whether this claim is technically correct. The number of parameters would depend on $N$ and $M$, and we would need to set it as the largest number we expect in the future. And I believe the mapping of orders/warehouses to these $N$ and $M$ should be done carefully. For example, if we simply assign 1, 2, ..., |I| as order indices, then embedding vectors for large indices close to $N$ wouldn't be trained well. What do authors do to make sure embedding vectors for all $N$ and $M$ indices are trained well?

I am curious how often does the model output a feasible solution. Although Section 4.4 describes how the model sequentially samples predictions to output feasible action, it may run into "dead end" which no more feasible output exists. The same question goes to other baseline algorithms as well. Unless all models always produce feasible solutions, it'd be useful to have additional column on the fraction of feasible solutions.

Minor comments:
* Figure 3 doesn't really tell anything, (a) and (b) look just the same. I suggest authors to describe exactly what we should look into in this plot.
* The notation $N$ is overloaded for the maximum number of orders as well as the layer of graph encoder.

**Summary Of The Paper:**

Authors propose to solve an order fulfillment problem with imitation learning backed by GNNs. The specific version of the fulfillment problem is nontrivial: discussed has a hierarchy in order (an order can be decomposed into sub-orders), and items become forbidden over time. SCIP package is used as the expert policy, and authors employ behavior cloning. The specific model they use is a Graph Attention Transformer on the tripartite graph of orders, items, and warehouses. Compared to a reasonable heuristic, Pointer Network, and GAT which ignores edge features, the propose method yields much lower inference latency.

**Summary Of The Review:**

The paper employs a methodically sound approach on a practically important problem. However, important details of the experiments as well as reasonable baselines are missing.

---

### Decision · Program_Chairs · 2022-01-20

**Decision:**

Reject

**Comment:**

A GNN model is developed for the supervised, real-time learning of optimal solutions for an order-fulfillment problem. GNNs with fast forward computations are naturally one good choice given the real-time nature of the problem.

While the complexity of the problem and formulation were generally appreciated by the referees, there were major concerns about the experimental setup, datasets, technical claims, sample complexity, and suitability for ICLR. Overall, the paper does not seem ready for publication in ICLR, and the authors are encouraged to consider and work on the reviews carefully.